# ON PROGRESSIVE SHARPENING, FLAT MINIMA AND GENERALISATION

## ABSTRACT

We present a new approach to understanding the relationship between loss curvature and input-output model behaviour in deep learning. Specifically, we use existing empirical analyses of the spectrum of deep network loss Hessians to ground an ansatz tying together the loss Hessian and the input-output Jacobian over training samples during the training of deep neural networks. We then prove a series of theoretical results which quantify the degree to which the input-output Jacobian of a model approximates its Lipschitz norm over a data distribution, and deduce a novel generalisation bound in terms of the empirical Jacobian. We use our ansatz, together with our theoretical results, to give a new account of the recently observed progressive sharpening phenomenon, as well as the generalisation properties of flat minima. Experimental evidence is provided to validate our claims.

## 1 INTRODUCTION

In this paper, we attempt to clarify how the curvature of the loss landscape of a deep neural network is related to the input-output behaviour of the model. Via a single mechanism, our Ansatz 3.1, we offer new explanations of both the as-yet unexplained progressive sharpening phenomenon Cohen et al. (2021) observed in the training of deep neural networks, and the long-speculative relationship between loss curvature and generalisation Hochreiter & Schmidhuber (1997); Keskar et al. (2017); Chaudhari et al. (2017); Foret et al. (2021).

The mechanism we propose in Ansatz 3.1 to mediate the relationship between loss curvature and input-output model behaviour is the input-output Jacobian of the model over the training sample, which is already understood to play a role in determining generalisation Drucker & Le Cun (1992); Bartlett et al. (2017); Neyshabur et al. (2017); Hoffman et al. (2019); Bubeck & Sellke (2021); Gouk et al. (2021); Novak et al. (2018); Ma & Ying (2021). Our proposal is based on empirical observations made of the eigenspectrum of the Hessian of deep neural networks Papyan (2018; 2019); Ghorbani et al. (2019), whose outliers can be attributed to a summand of the Hessian known as the Gauss-Newton matrix. Crucially, the Gauss-Newton matrix is second-order only in the cost function, and solely first-order in the network layers.

The Gauss-Newton matrix is a Gram matrix (i.e. a product $A^T A$ for some matrix $A$), and its conjugate $AA^T$, closely related to the tangent kernel identified in Jacot et al. (2018), has the same nonzero eigenvalues. Expanding $AA^T$ reveals that it is determined in part by composite input-output layer Jacobians. Thus, insofar as the outlying Hessian eigenvalues are determined by those of the Gauss-Newton matrix, and insofar as input-output layer Jacobians determine the spectrum of the Gauss-Newton matrix via its isospectrality to its conjugate, one expects the largest singular values of the loss Hessian to be closely related to those of the model's input-output Jacobian.

Our contributions in this paper are as follows.

1. Based on previous empirical work identifying outlying loss Hessian eigenvalues with those of the Gauss-Newton matrix, we propose an ansatz: that under certain conditions, the largest eigenvalues of the loss Hessian control the growth of the largest singular values of the model's input-output Jacobian. We focus in particular on the *largest* eigenvalue of the Hessian (the *sharpness*) and the *largest* singular value of the Jacobian (its *spectral norm*, which we abbreviate to simply *norm* in what follows).

2. In Section 4 we provide theorems which quantify the extent to which the maximum input-output Jacobian spectral norm of a model over a training set will approximate the Lipschitz norm of the model over the underlying data distribution during training for datasets of practical relevance, including those generated by a generative adversarial network (GAN) or implicit neural function.

3. In Section 5, we provide a theorem which gives a data-dependent, high probability lower bound on the empirical Jacobian norm of a model that grows during any effective training procedure. We combine this result with Ansatz 3.1 to give a new account of progressive sharpening, which enables us to change the severity of sharpening by scaling inputs and outputs. We report on classification experiments that validate our account.

4. In Section 6, we provide a novel bound on the generalisation gap of the model in terms of the empirical Jacobian norm. Synthesising this result with Ansatz 3.1, we argue that low loss curvature implies good generalisation only insofar as low loss curvature implies small empirical Jacobian norm. We report on experiments measuring the effect of hyperparameters such as learning rate and batch size on loss sharpness, Jacobian norm and generalisation, revealing the validity of our explanation. Finally, we present the results of experiments measuring the loss sharpness, Jacobian norm and generalisation gap of networks trained with a variety of regularisation measures, revealing in all cases the superior correlation of Jacobian norm and generalisation when compared with loss sharpness, whether or not the regularisation technique targets loss sharpness.

## 2 RELATED WORK

**Flatness, Jacobians and generalisation:** The position that flatter minima generalise better dates back to Hochreiter & Schmidhuber (1997). It has since become a staple concept in the toolbox of deep learning practitioners Keskar et al. (2017); Chaudhari et al. (2017); Foret et al. (2021); Chen et al. (2020), with its incorporation into training schemes yielding state-of-the-art performance in many tasks. Its effectiveness has motivated the study of the effect that learning hyperparameters such as learning rate and batch size have on the sharpness of minima to which the algorithm will converge Wu et al. (2018); Zhu et al. (2019); Mulayoff et al. (2021). The hypothesis has received substantial criticism. In Dinh et al. (2017) it is shown that flatness is not *necessary* for good generalisation in ReLU models, since such models are invariant to scaling symmetries of the parameters which arbitrarily sharpen their loss landscapes. This observation has motivated the consideration of scale-invariant measures of loss sharpness Lyu et al. (2022); Jang et al. (2022). In Granziol (2020), the *sufficiency* of loss flatness for good generalisation is disputed by showing that models trained with the cross-entropy cost generalise better with weight decay than without, despite the former ending up in sharper minima. Sufficiency of flatness for generalisation is further challenged for Gaussian-activated networks on regression tasks in Ramasinghe et al. (2023).

Relatively little work has looked into the mechanism underlying the relationship between loss curvature and generalisation. PAC-Bayes bounds Dziugaite & Roy (2017); Foret et al. (2021); F. He and T. Liu and D. Tao (2019) provide some theoretical support for the hypothesis that loss flatness is sufficient for good generalisation, however this hypothesis, taken unconditionally, is known to be false empirically Granziol (2020); Ramasinghe et al. (2023). The same is true of Petzka et al. (2021), which cleverly bounds the generalisation gap by a rescaled loss Hessian trace. Despite advocating for the empirical Jacobian as the link between loss curvature and generalisation as we do, the insightful papers Ma & Ying (2021); Gamba et al. (2023) consider the relationship only at a critical point of the square cost, and in particular cannot offer explanations for progressive sharpening or the generalisation benefits of training with an only an *initially* large learning rate that is gradually decayed, as our account can. Lee et al. (2023) advocates empirically for the parameter-output Jacobian of the model as the mediating link between loss sharpness and generalisation, but provide no theoretical indication for why this should be the case. We provide evidence in Appendix D indicating that our proposal may be able to fill this gap in Lee et al. (2023).

Finally, we note that among other generalisation studies in the literature, ours is most closely related to Ma & Ying (2021) which exploits properties of the data distribution to prove a generalisation bound. Our bound is also related to Bartlett et al. (2017); Bubeck & Sellke (2021); Muthukumar & Sulam (2023), which recognise sensitivity to input perturbations as key to generalisation. Like

Wei & Ma (2019), our work in addition formalises the intuitive idea that the empirical Jacobian norm regulates generalisation Drucker & Le Cun (1992); Hoffman et al. (2019); Novak et al. (2018). Further study of our ansatz in relation to the edge of stability phenomenon Cohen et al. (2021) may be of utility to algorithmic stability approaches to generalisation Bousquet & Elisseeff (2002); Hardt et al. (2016); Charles & Papailiopoulos (2018); Kuzborskij & Lampert (2018); Bassily et al. (2020).

**Progressive sharpening and edge of stability:** The effect of learning rate $\eta$ on loss sharpness has been understood to some extent for several years Wu et al. (2018). In Cohen et al. (2021), this relationship was attended to for deep networks with a rigorous empirical study which showed that, after an initial period of sharpness increase during training called *progressive sharpening*, the sharpness increase halted at around $2/\eta$ and oscillated there while loss continued to decrease non-monotonically, a phase called *edge of stability*. These phenomena show that the typical assumptions made in theoretical work, namely that the learning rate is always smaller than twice the reciprocal of the sharpness Allen-Zhu et al. (2019); Du et al. (2019b;a); MacDonald et al. (2022), do not hold in practice. Significant work has since been conducted to understand these phenomena Lyu et al. (2022); Lee & Jang (2023); Arora et al. (2022); Wang et al. (2022); Zhu et al. (2023); Ahn et al. (2022), with Damian et al. (2023) showing in particular that edge of stability is a universal phenomenon arising from a third-order stabilising effect that must be taken into account when the sharpness grows above $2/\eta$. In contrast, progressive sharpening is not universal; it is primarily observed only in genuinely deep neural networks. Although it has been correlated to growth in the norm of the output layer Wang et al. (2022), the cause of progressive sharpening has so far remained mysterious. The mechanism we propose is the first account of which we are aware for a *cause* of progressive sharpening.

## 3 BACKGROUND AND PAPER OUTLINE

### 3.1 OUTLIERS IN THE SPECTRUM

We follow the formal framework introduced in MacDonald et al. (2022), which allows us to treat all neural network layers on a common footing. Specifically, we consider a multilayer parameterised system $\{f_i : \mathbb{R}^{p_i} \times \mathbb{R}^{d_{i-1}} \to \mathbb{R}^{d_i}\}_{i=1}^{L}$ (the layers), with a data matrix $X \in \mathbb{R}^{d_0 \times N}$ consisting of $N$ data vectors in $\mathbb{R}^{d_0}$. We denote by $F : \mathbb{R}^{p_1 + \cdots + p_L} \to \mathbb{R}^{d_L \times N}$ the associated parameter-function map defined by

$$F_X(\theta_1, \ldots, \theta_L)_i := f_L(\theta_L) \circ \cdots \circ f_1(\theta_1)(X)_i \in \mathbb{R}^{d_L} \qquad i = 1, \ldots, N. \tag{1}$$

Given a convex cost function $c : \mathbb{R}^{d_L} \times \mathbb{R}^{d_L} \to \mathbb{R}$ and a target matrix $Y \in \mathbb{R}^{d_L \times N}$, we consider the associated loss

$$\ell(\vec{\theta}) := \gamma_Y \circ F_X(\vec{\theta}) = \frac{1}{N} \sum_{i=1}^{N} c\big(F_X(\vec{\theta})_i, Y_i\big), \tag{2}$$

where $\gamma_Y : \mathbb{R}^{d_L \times N} \to \mathbb{R}$ is defined by $\gamma_Y(Z) := N^{-1} \sum_{i=1}^{N} c(Z_i, Y_i)$.

Using the chain rule and the product rule, one observes that the Hessian $D^2\ell$ of $\ell$ admits the decomposition

$$D^2\ell = DF_X^T \, D^2\gamma_Y \, DF_X + D\gamma_Y \, D^2 F_X. \tag{3}$$

The first of these terms, often called the *Gauss-Newton matrix*, is positive-semidefinite by convexity of $\gamma_Y$. It has been demonstrated empirically in a vast number of practical settings that the largest, outlying eigenvalues of the Hessian throughout training correlate closely with those of the Gauss-Newton matrix Papyan (2018; 2019); Cohen et al. (2021). In attempting to understand the relationship between loss sharpness and model behaviour, therefore, empirical evidence invites us to devote special attention to the Gauss-Newton matrix. **In what follows we assume based on this evidence that the outlying eigenvalues of the Gauss-Newton matrix *determine* those of the loss Hessian**.

### 3.2 THE GAUSS-NEWTON MATRIX AND INPUT-OUTPUT JACOBIANS

Letting $C$ denote the square root $(D^2\gamma_Y)^{\frac{1}{2}}$, we see that the Gauss-Newton matrix $DF_X^T \, C^2 \, DF_X$ has the same nonzero eigenvalues as its conjugate matrix $C \, DF_X \, DF_X^T \, C^T$, which is closely related to the *tangent kernel* $DF_X \, DF_X^T$ identified in Jacot et al. (2018). Letting $Jf_l$ and $Df_l$ denote the

input-output and parameter derivatives of a layer $f_l : \mathbb{R}^p \times \mathbb{R}^{d_{l-1}} \to \mathbb{R}^{d_l}$ respectively, one sees that

$$C \, DF_X \, DF_X^T \, C^T = C \left( \sum_{l=1}^{L} \left( Jf_L \cdots Jf_{l+1} \, Df_l \, Df_l^T \, Jf_{l+1}^T \cdots Jf_L^T \right) \right) C^T \tag{4}$$

is a sum of positive-semidefinite matrices. Each summand is determined in large part by the composites of input-output layer Jacobians. Note, however, that the input-output Jacobian of the *first* layer does not appear; only its parameter derivative does.

It is clear then that insofar as the largest eigenvalues of the Hessian are determined by the Gauss-Newton matrix, these eigenvalues are determined in part by the input-output Jacobians of all layers following the first. We judge the following ansatz to be intuitively clear from careful consideration of Equation (4).

**Ansatz 3.1.** *Under certain conditions, an increase in the magnitude of the input-output Jacobian of a deep neural network will cause an increase in the loss sharpness. Conversely, a decrease in the sharpness will cause a decrease in the magnitude of the input-output Jacobian.*

Note that Ma & Ying (2021) already contains a rigorous proof of Ansatz 3.1 under special conditions: namely at critical points of the square cost. This restricted setting is not sufficient for our purposes, which require us to invoke Ansatz 3.1 *throughout training* and most frequently with the *cross-entropy cost function*, where Ma & Ying (2021) does not apply.

While Ansatz 3.1 is the basis for our explanations of both progressive sharpening and the good generalisation of flat minima, *we make no claim that Ansatz 3.1 holds unconditionally*. Indeed, we argue that the already-observed contingency of these phenomena derive precisely from the contingency of Anstaz 3.1. Specifically, the relationship proposed in Ansatz 3.1 is *necessarily* mediated by the following factors present in Equation (4): (1) the square root $C$ of the second derivative of the cost function; (2) the presence of the parameter derivatives $Df_l$; (3) the complete absence of the Jacobian of the first layer. In all cases where Ansatz 3.1 appears not to apply (Figure 2, Appendix C), we are able to account for its inadequacy in terms of these mediating factors using Equation (4) and further measurements. Moreover, we are able to to explain all known exceptions to the "rules" of the superior generalisation of flat minima Granziol (2020); Ramasinghe et al. (2023) and of progressive sharpening Cohen et al. (2021) as consequences of unusual behaviour in these mediating factors. Ours is the only account we are aware of for these phenomena which can make this claim.

## 4  GENERAL THEORY

All of what follows is motivated by the following idea: the Lipschitz norm of a differentiable function is upper-bounded by the supremum, over the relevant data distribution, of the spectral norm of its Jacobian. Intuitively, this supremum will itself be approximated by the maximum Jacobian norm over a finite sample of points from the distribution, which by Ansatz 3.1 can be expected to relate to the loss Hessian of the model. It is this intuition we seek to formalise and, and whose practical relevance we seek to justify, in this section.

In what follows, we will use $\mathbb{R}_{>0}$ to denote the positive real numbers. We will use $\mathbb{P}$ to denote a probability measure on $\mathbb{R}^d$, whose support $\text{supp}(\mathbb{P})$ we assume to be a metric space with metric inherited from $\mathbb{R}^d$. Given a Lipschitz function $g : \mathbb{R}^d \to \mathbb{R}^{d'}$, we use $\|g\|_{Lip,\mathbb{P}}$ to denote the Lipschitz norm of $g|_{\text{supp}(\mathbb{P})}$. Note the dual meaning of $\|g\|_2$: when $g$ is *vector*-valued, it refers to the Euclidean norm, while when $g$ is *operator*-valued (e.g., when $g = Jf$ is a Jacobian), $\|g\|_2$ refers to the spectral norm. We will frequently invoke pointwise evaluation of the derivative of Lipschitz functions, and in doing so always rely on the fact that our evaluations are probabilistic and Lipschitz functions are differentiable almost everywhere. We will use $B(x, \delta) \subset \mathbb{R}^d$ to denote the *closed* Euclidean ball of radius $\delta$ centered $x$. Finally, given a locally bounded (but not necessarily continuous) function $g : \mathbb{R}^d \to \mathbb{R}^{d'}$ and a compact set $S \subset \mathbb{R}^d$, we define the *local variation of $g$ over $S$* by

$$V_S(g) := \sup_{x,y \in S} \|g(x) - g(y)\|_2. \tag{5}$$

Thus, for instance, if $g$ is Lipschitz and $S$ is a ball of radius $\delta$, then $V_S(g) \le 2\delta \|g|_S\|_{Lip}$. All proofs are deferred to the appendix.

We begin by specifying the data distributions with which we will be concerned.

**Definition 4.1.** Let $\delta : \mathbb{N} \times (0, 1) \to \mathbb{R}_{>0}$ be a function such that for each $\epsilon \in (0, 1)$, the function $N \mapsto \delta(N, \epsilon)$ is decreasing and vanishes as $N \to \infty$. We say that a distribution $\mathbb{P}$ is $\delta$-**good** if for every $N \in \mathbb{N}$ and $\epsilon \in (0, 1)$, with probability at least $1 - \epsilon$ over i.i.d. samples $(x_1, \ldots, x_N)$ from $\mathbb{P}$, one has $\text{supp}(\mathbb{P}) \subset \bigcup_{i=1}^{N} B(x_i, \delta(N, \epsilon))$.

A similar assumption on the data distribution is adopted in Ma & Ying (2021), however no *proof* is given in Ma & Ying (2021) that such an assumption is characteristic of data distributions of practical interest. Our first theorem, derived from (Reznikov & Saff, 2016, Theorem 2.1), is that many data distributions of practical interest in deep learning are indeed examples of Definition 4.1.

**Theorem 4.2.** *Suppose that $\mathbb{P}$ is the normalised Riemannian volume measure of a compact, connected, embedded, $d$-dimensional submanifold of Euclidean space (possibly with boundary and corners). Then there exists $\delta : \mathbb{N} \times (0, 1) \to \mathbb{R}_{>0}$, with $\delta(N, \epsilon) = O\big((\log(N\epsilon^{-1})N^{-1})^{\frac{1}{d}}\big)$, such that $\mathbb{P}$ is $\delta$-good.*

*In particular, the uniform distribution on the unit hypercube $[0, 1]^d$ in $\mathbb{R}^d$ is $\delta_{[0,1]^d}$-good with*

$$\delta_{[0,1]^d}(N, \epsilon) = \frac{4\,\Gamma(\frac{d}{2} + 1)^{\frac{1}{d}}}{\sqrt{\pi}} \left( \frac{\log(N\epsilon^{-1})}{N} \right)^{\frac{1}{d}}, \tag{6}$$

*and, for $0 < \gamma < 1$ and $N$ sufficiently large, the uniform distribution on the $d$-dimensional unit hypersphere $S^d \subset \mathbb{R}^{d+1}$ is $\delta_{S^d}$-good with*

$$\delta_{S^d}(N, \epsilon) = 2^{1+\frac{1}{d}}(1 - \gamma)^{-\frac{1}{2}} \left( \frac{\log(N\epsilon^{-1})}{N} \right)^{\frac{1}{d}}. \tag{7}$$

*Moreover, if $\mathbb{P}$ is any $\delta$-good distribution, then the pushforward of $\mathbb{P}$ by any Lipschitz function $g$ is $\|g\|_{Lip,\mathbb{P}}\delta$-good.*

Theorem 4.2 says in particular that the distribution generated by any GAN whose latent space is the uniform distribution on a hypercube or on a sphere is an example of a good distribution, as is any distribution generated by an implicit neural function Sitzmann et al. (2020). Since a high dimensional isotropic Gaussian is close to a uniform distribution on a sphere, which is good by Theorem 4.2, we believe it likely that our theory can be strengthened to include Gaussian distributions, however we have not attempted to prove this and leave it to future work.

Our next theorem formalises our intuition that the maximum Jacobian is an approximate upper bound on the Lipschitz constant of a model throughout training. Its proof is implicit in the proofs of both of our following theorems which characterise progressive sharpening and generalisation in terms of the empirical Jacobian norm.

**Theorem 4.3.** *Suppose that $\mathbb{P}$ is $\delta$-good, and let $N \in \mathbb{N}$ and $0 < \epsilon < 1$. Then with probability at least $1 - \epsilon$ over i.i.d. samples $(x_1, \ldots, x_N)$ from $\mathbb{P}$, for any Lipschitz function $f : \mathbb{R}^d \to \mathbb{R}^{d'}$, one has:*

$$\|f\|_{Lip,\mathbb{P}} \leq \max_i \big( \|Jf(x_i)\|_2 + V_{B(x_i, \delta(N, \epsilon))}(Jf) \big). \tag{8}$$

## 5 PROGRESSIVE SHARPENING

In this section we give our account of progressive sharpening. We will assume that training of a model $f : \mathbb{R}^d \to \mathbb{R}^{d'}$ is undertaken using a cost function $c : \mathbb{R}^{d'} \times \mathbb{R}^{d'} \to \mathbb{R}$, whose global minimum we assume to be zero, and for which we assume that there is $\alpha > 0$ such that $c(z_1, z_2) \geq \alpha\|z_1 - z_2\|_2^2$ for all $z_1, z_2 \in \mathbb{R}^{d'}$. This is clearly always the case for the square cost; by Pinsker's inequality, it also holds for the Kullback-Leibler divergence on softmax outputs, and hence also for the cross-entropy cost provided one subtracts label entropy.

**Theorem 5.1.** *Suppose that $\mathbb{P}$ is $\delta$-good, and let $N \in \mathbb{N}$ and $0 < \epsilon < 1$. Let $f^* : \text{supp}(\mathbb{P}) \to \mathbb{R}^{d'}$ be a target function and fix a real number $\ell > 0$. Then with probability at least $1 - \epsilon$ over i.i.d. samples $(x_1, \ldots, x_N)$ drawn from $\mathbb{P}$, for any Lipschitz function $f : \mathbb{R}^d \to \mathbb{R}^{d'}$ satisfying $c(f(x_i), f^*(x_i)) \leq \ell^2\alpha$ for all $i$, one has*

$$\max_i \|Jf(x_i)\|_2 \geq \max_{i \neq j} \frac{\|f^*(x_i) - f^*(x_j)\|_2 - 2\ell}{\|x_i - x_j\|_2} - \max_i V_{B(x_i, \delta(N, \epsilon))}(Jf). \tag{9}$$

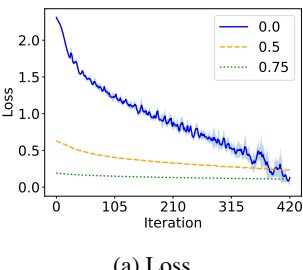 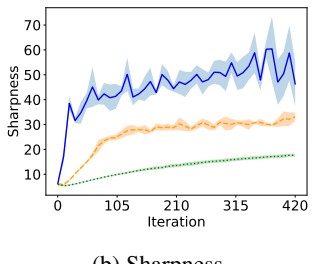 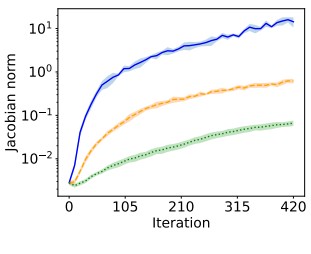

(a) Loss                (b) Sharpness                (c) Jacobian norm (log $y$ axis)

Figure 1: Plots of cross entropy loss (with label entropies subtracted), loss sharpness and softmaxed-model Jacobian norm with varying degrees of label smoothing, for VGG11 trained with gradient descent on CIFAR10, with a learning rate of 0.08. Increasing label smoothing (indicated by line style) brings targets closer together, meaning less growth is necessary in the Jacobian norm to reduce loss. This coincides with less sharpening of the Hessian during training, in line with Ansatz 3.1. Similar plots at different learning rates, and with ResNet18, are in Appendix B.1. Note log $y$ axis on Jacobian norm for ease of distinction.

**Progressive sharpening:** Theorem 4.3 tells us that any training procedure that reduces the loss over all data points will thus also increase the sample-maximum Jacobian norm from a low starting point. Invoking Ansatz 3.1, this increase in the sample-maximum Jacobian norm can be expected to cause a corresponding increase in the magnitude of the loss Hessian (see Appendix A.1): progressive sharpening.

Theorem 5.1 thus tells us that sharpening can be made more or less severe by scaling the distances between target values: indeed this is what we observe (Figure 1, see also Appendix B.1). One might also expect that for non-batch-normalised networks, scaling the inputs closer together would increase sharpening too. However, this is not the case, due to the mediating factors in Ansatz 3.1. Specifically, while it is true that such scaling increases the growth rate of the Jacobian norm, this increase in Jacobian norm does *not* necessarily increase sharpening, since it coincides with a *decrease* in the magnitude of the parameter derivatives, which are key factors in relating the model Jacobian to the loss Hessian (see Appendix C.2).

**Edge of stability:** Although the edge of stability mechanism explicated in Damian et al. (2023) can be expected to put some downward pressure on the model Jacobian, due to the presence of the mediating factors discussed following Ansatz 3.1 it *need not* cause the Jacobian to *plateau* in the same way that loss sharpness does. Nonetheless, this downward pressure on Jacobian is important: it is to this that we attribute the better generalisation of models trained with large learning rate, even if that learning rate is decayed towards the end of training. We validate this empirically in the next section and Appendix D.

**Wide networks and linear models:** It has been observed empirically that wide networks exhibit less severe sharpening, and linear (kernel) models exhibit none at all Cohen et al. (2021). It might be thought, since these models nonetheless must increase in Jacobian norm during training by Theorem 5.1, that such models are therefore counterexamples to our explanation. They are in fact consistent: Theorem 5.1 only implies sharpening *insofar as Ansatz 3.1 holds*. Importantly, since the Jacobian of the first linear layer does not appear in Equation (4), an increase in the input-output Jacobian norm of a linear (kernel) model will not cause an increase in sharpness according to Ansatz 3.1. The same is true of wide networks, which approximate kernel models increasingly well as width is sent to infinity Lee et al. (2019); thus wide networks will also be expected to sharpen less severely during training, as has been observed empirically.

## 6    FLAT MINIMA AND GENERALISATION

We argue in this section that loss flatness implies good generalisation only via encouraging smaller Jacobian norm, through Ansatz 3.1. Indeed, our final theorem, whose proof is inspired by that of (Ma & Ying, 2021, Theorem 6), assures us rigorously that on models that fit training data, sufficiently small

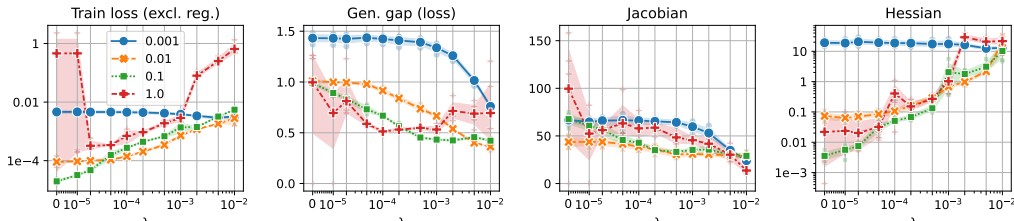

Figure 2: The impact of weight decay ($x$ axis) on Jacobian norm, sharpness and generalisation gap when training ResNet18 with cross-entropy loss at the end of training (90 epochs). Line style indicates learning rate. Without weight decay, SGD finds a solution with near-zero loss and vanishing sharpness, but for which Jacobian norm and generalisation gap are relatively large.

empirical Jacobian norm and sufficiently large sample size suffice to guarantee good generalisation, as has long been suspected in the literature Drucker & Le Cun (1992).

**Theorem 6.1.** *Let $\mathbb{P}$ be $\delta$-good and let $f^* : \text{supp}(\mathbb{P}) \to \mathbb{R}^{d'}$ be a target function, which we assume to be Lipschitz. Let $N \in \mathbb{N}$ and let $0 < \epsilon < 1$. Then with probability at least $1 - \epsilon$ over all i.i.d. samples $(x_1, \ldots, x_N)$ from $\mathbb{P}$, any Lipschitz function $f : \mathbb{R}^d \to \mathbb{R}^{d'}$ which coincides with $f^*$ on $(x_1, \ldots, x_N)$, satisfies:*

$$\mathbb{E}_{x \sim \mathbb{P}} \|f(x) - f^*(x)\|_2 \leq \delta(N, \epsilon) \big( \|f^*\|_{Lip,\mathbb{P}} + \max_i \big( \|Jf(x_i)\|_2 + V_{B(x_i, \delta(N,\epsilon))}(Jf) \big) \big). \quad (10)$$

Our proof of Theorem 6.1 is similar to that of (Ma & Ying, 2021, Theorem 6). Our bound is an improvement on that of Ma & Ying (2021) in having better decay in $N$. Additionally, in bounding in terms of the empirical Jacobian norm at convergence instead of training hyperparameters at convergence, our bound is sensitive to implicit Jacobian regularisation throughout training in a way that the bound of Ma & Ying (2021) is not. Our bound is thus in principle sensitive to the superior generalisation of networks trained with an *initially* high learning rate that is gradually decayed to a small learning rate (such networks experience more implicit Jacobian regularisation at the edge of stability), over networks trained with a small learning rate from initialisation. In contrast, the bound of Ma & Ying (2021) is not sensitive to this distinction.

It is worth nothing that our bound exhibits *slower* $O\big((\log(N)N^{-1})^{\frac{1}{d}}\big)$ decay in $N$, where $d$ is the intrinsic dimension of the data distribution, than the more common $O(N^{-\frac{1}{2}})$ rates in the literature. The reason for this is that all bounds of which we are aware in the literature, with the exception of (Ma & Ying, 2021, Theorem 6), derive from consideration of *hypothesis complexity*, rather than *data complexity* as we and Ma & Ying (2021) consider. Although the method we adopt implies this worse decay in $N$, our method has the advantage of not needing to invoke the large hypothesis complexity terms, such as Rademacher complexity and KL-divergence, that make bounds based on such terms so loose when applied to deep learning Zhang et al. (2017). Moreover, since standard datasets in deep learning are intrinsically low dimensional Miyato et al. (2018), the $O\big((\log(N)N^{-1})^{\frac{1}{d}}\big)$ rate in our bounds can still be expected to be nontrivial in practice.

In what follows we present experimental results measuring the generalisation gap (absolute value of train loss minus test loss), sharpness and Jacobian norm at the end of training with varying degrees of various regularisation techniques. Our results confirm that while loss sharpness is neither necessary nor sufficient for good generalisation, empirical Jacobian norm consistently correlates better with generalisation than does sharpness for all regularisation measures we studied. Moreover, whenever loss sharpness *does* correlate with generalisation, this correlation tends to go hand-in-hand with smaller Jacobian norm, as predicted by Ansatz 3.1. Note that the Jacobians we plot were all computed in evaluation mode, meaning all BN layers had their statistics fixed: that the relationship with the train mode loss Hessian can still be expected to hold is justified in Appendix A.1.

**Cross-entropy and weight decay:** It was observed in Granziol (2020) that when training with the cross-entropy cost, networks trained with weight decay generalised better than those trained without, despite converging to sharper minima. We confirm this observation in Figure 2. Using Ansatz 3.1 and Theorem 6.1, we are able to provide a new explanation for this fact. For the cross-entropy cost, both terms $D^2\gamma_Y$ and $D\gamma_Y$ appearing in the Hessian (Equation (3)) vanish at infinity in parameter space.

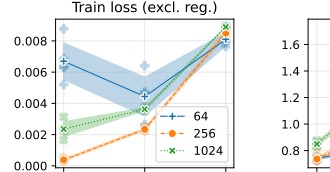 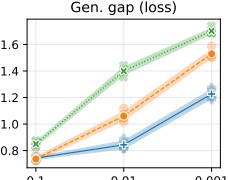 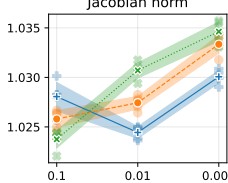 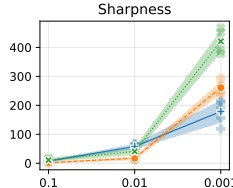

Figure 3: The impact of differing (constant) learning rates ($x$ axis) at end of training with ResNet18 on CIFAR10 trained with SGD (five trials). Line style indicates batch size. In line with conventional wisdom, larger learning rates typically have smaller sharpness and generalise better. The downward pressure on sharpness and generalisation gap correlates with smaller Jacobian norms, as anticipated by Ansatz 3.1 and Theorem 6.1. Models trained with weight decay (0.0001).

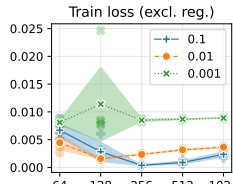 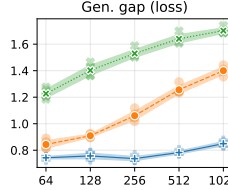 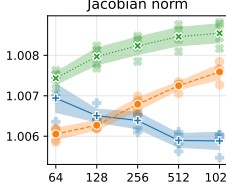 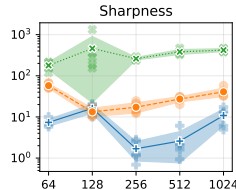

Figure 4: The impact of differing batch sizes ($x$ axis) at the end of training with ResNet18 on CIFAR10 trained with SGD (five trials). Line style indicates (constant) learning rate. At least for the smaller learning rates, increasing batch size typically increases sharpness, Jacobian norm and generalisation gap in line with Theorem 6.1. At the largest learning rate, the relationship between Jacobian norm and generalisation gap is the inverse of what is expected, suggesting that larger learning rates tend to increase the local variation of the model's Jacobian, causing it to underestimate the Lipschitz constant of the model (Theorem 6.1). Note log scale on $y$ axis in Sharpness plot for ease of distinction.

By preventing convergence of the parameter vector to infinity in parameter space, weight-decay therefore encourages convergence to a sharper minimum than would otherwise be the case. However, since the Frobenius norm of a matrix is an upper bound on the spectral norm, weight decay also implicitly regularises the Jacobian of the model, leading to better generalisation by Theorem 6.1.

Since training with weight decay is the norm in practice, and since any correlations between *vanishing* Hessian and generalisation will likely be unreliable, we used weight decay in all of what follows.

**Learning rate and batch size:** It is commonly understood that larger learning rates and smaller batch sizes serve to encourage convergence to flatter minima Keskar et al. (2017); Wu et al. (2018). By Theorem 6.1 and Ansatz 3.1, such hyperparameter choice can be expected to lead to better generalisation. We report on experiments testing this idea in Figures 3, 4 (ResNet18 on CIFAR10) and Appendix B.2 (ResNet18 and VGG11 on CIFAR10 and CIFAR100). The results are mostly consistent with expectations, with the exception of Jacobian norm being reduced with larger batch size at the highest learning rate while generalisation gap increases (Figure 4). The phenomenon appears to be data-agnostic (the same occurs with ResNet18 on CIFAR100), but architecture-dependent (we do not observe this problem with VGG11 on either CIFAR10 or CIFAR100). This is not necessarily inconsistent with Theorem 6.1, which allows for the empirical Jacobian norm to be a poor estimator of the Lipschitz constant of the model via the local variation in the model's Jacobian. Our results indicate that this local variation term is increased during (impractical) large batch training of skip-connected architectures using a constant high learning rate.

**Other regularisation techniques:** We also test to see the extent that other common regularisation techniques, including label smoothing, data augmentation, mixup Zhang et al. (2018) and sharpness-aware minimisation (SAM) Foret et al. (2021) regularise Hessian and Jacobian to result in better generalisation. We find that while Jacobian norm is regularised at least initially across all techniques, only in SAM is sharpness regularised. This validates our proposal that Jacobian norm is a key mediating factor between sharpness and generalisation, as well as the position of Dinh et al. (2017)

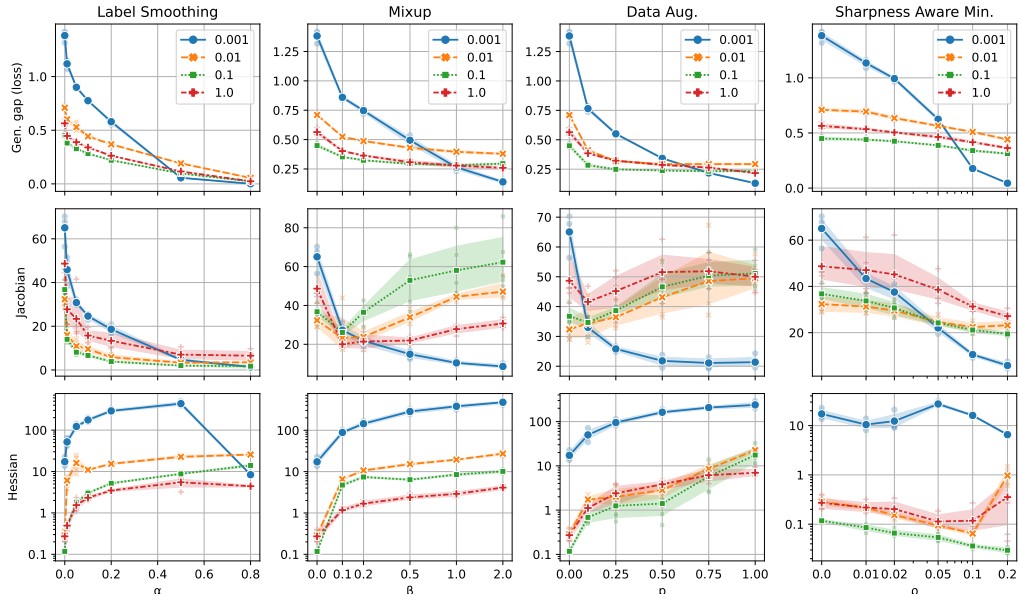

Figure 5: The impact of different regularisation strategies ($x$ axis) when training a ResNet18 model with batch-norm on CIFAR10 (five trials). The norm of the input-output Jacobian is much better correlated with the generalisation gap than that of the Hessian; all regularisation strategies were effective in controlling the Jacobian norm without necessarily controlling the sharpness. Line style indicates initial learning rate. Besides the regularisation strategy being studied, all experiments include weight decay (0.0005). Note that the decorrelation between Jacobian and generalisation for Mixup and Data Aug as $x$-axis parameter is increased is to be expected. In both cases, we measure the Jacobian norm over *only the pure training set*, which makes up less of a percentage of the total training set and is therefore subject to less implicit regularisation during training as the $x$-axis parameter is increased. Refer to the appendix for extended results.

that flatness is not necessary for good generalisation. Note that at least some of the gradual increase in Jacobian norm for mixup and data augmentation as the $x$-axis parameter is increased is to be expected due to our measurement scheme: see the caption in Figure 5. See Appendix B.3 for experimental details and similar results for VGG11 and CIFAR100.

## 7 DISCUSSION

One limitation of our work is that Ansatz 3.1 is not mathematically rigorous. A close theoretical analysis would ideally provide rigorous conditions under which one can upper bound the model Jacobian norm in terms of loss sharpness *throughout training*. These conditions would need to be compatible with the empirical counterexmaples we give in Figure 2 and Appendix C.

A second limitation of our work is that we do not numerically evaluate our generalisation bound. Numerical evaluation of the bound would require both an estimate of the Lipschitz constant of a GAN which has generated the data, as well as the local variation of the Jacobian of the model defined in Equation (5). While existing techniques may be able to aid the first of these Fazlyab et al. (2019), we are unaware of any work studying the second, which is outside the scope of this paper. We leave this important evaluation to future work.

## 8 CONCLUSION

We proposed a new relationship between the loss Hessian of a deep neural network and the input-output Jacobian of the model. We proved several theorems allowing us to leverage this relationship in providing new explanations for progressive sharpening and the link between loss geometry and generalisation. An experimental study was conducted which validates our proposal.

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
