# OpenReview forum: "On progressive sharpening, flat minima and generalisation"
_ICLR.cc/2024/Conference — Submitted to ICLR 2024_

### Official Review · Reviewer_dFnj · 2023-10-30

**Soundness:** 3 good
**Presentation:** 3 good
**Contribution:** 3 good
**Rating:** 8
**Confidence:** 2

**Summary:**

The authors consider a decomposition of the Hessian of a convex loss for neural networks that yields the Gauss-Newton matrix as one of its terms which in turn can be decomposed as a sum over composites of individual input-output Jacobians of the neural network layers. The authors argue that, based on empirical evidence, the large (outlying) eigenvalues of the Gauss-Newton matrix determine those of the Hessian (curvature) and the spectrum of the Gauss-Newton matrix is in turn determined by the extreme singular values of the input-output Jacobians. So to understand the progressive sharpening phenomenon and better generalization with flat curvature it makes sense to analyze the input-output Jacobian norm. The authors’ theoretical contributions explain progressive sharpening, the connection between the Lipschitz norm of a neural network, its input-output Jacobian, and its generalization gap for the given input distribution assumptions. Through numerical experiments the authors also demonstrate how the Jacobian norm is correlated to sharpness and the generalization gap in practice and how different regularization approaches such as label smoothing, weight decay, sharpness aware minimization, data augmentation, learning rate changes etc. impact generalization gap, Jacobian norm and sharpness. The theoretical approach in the paper is compatible with the view that loss flatness does not generally imply generalization because of possible reparameterization (the Dinh et al reference).

**Strengths:**

- The theory in this paper is driven by empirical observations and explains effects that intuitively make sense and that have been observed by practitioners in a way that contributes to a deeper understanding about generalization.
- The empirical results in the paper are well chosen to demonstrate the various effects discussed and cover a large part of relevant regularization and hyperparameter dimensions

**Weaknesses:**

- The authors argue that their generalization bound (Theorem 6) is superior compared to other generalization bounds in the literature because it involves data complexity over hypothesis complexity. As a reason, the authors cite that datasets in standard deep learning are intrinsically low dimensional and hence the rate of their bound can be nontrivial in practice. But it does not become clear from the paper how the intrinsic dimensionality is estimated in practice and whether that means that it's possible to tightly estimate actual generalization gaps in practice based on their rate in practice (even for simple examples).

**Questions:**

- The main claims / contributions about progressive sharpening (Theorem 5.1) and generalization (Theorem 6.1) could be summarized or more clearly highlighted already in the abstract / introduction.
- For the paper to be more self-contained iIt would be helpful to clearly list and explain the empirical phenomena explained by the theory. E.g. the implicit regularization through higher initial learning rate / edge of stability phenomenon may not be familiar to all readers.

---

> ### Author Response · Authors · 2023-11-16
> **Thank you for your time and appreciation**
>
> **Weaknesses: generalisation bound practicalities**: We think it is too strong to say that our bound is *superior* to others in the literature: what we would instead say is that the two approaches (data complexity versus hypothesis complexity) each have their advantages and disadvantages. Our approach is insensitive to hypothesis complexity which makes bounds based on such approaches so loose, but suffers from a worse rate of decrease in the number of training points.
>
> One way that it may be possible to evaluate our bound in practice would be to generate the data from a GAN whose latent space is the uniform distribution on a hypersphere, or perhaps several GANs (one for each class) to take maximum advantage of the low-dimensionality of the data. If one could estimate the Lipschitz constant of the generators, as well as the local variation terms in our bound, then evaluation of our bound would be possible. However, we believe these evaluations would themselves require substantial additional research and are outside the scope of this paper.
>
> **Questions: clarification of contributions in abstract/introduction, and more self-contained narrative** Thank you for the recommendations. We agree that they are worth implementing and will consider how to fit them into the tight space limits we are constrained by.

---

### Official Review · Reviewer_7LFk · 2023-11-01

**Soundness:** 3 good
**Presentation:** 3 good
**Contribution:** 2 fair
**Rating:** 6
**Confidence:** 2

**Summary:**

This paper expands on previous work exploring the relationship between the curvature of the loss landscape (the Hessian eigenvalues) and the sensitivity of the input-output mapping of the neural network (the singular values of the input-output Jacobian). The key claim is that these two properties vary together in certain circumstances (Ansatz 3.1), and this assertion is then leveraged to study mechanisms underlying progressive sharpening and the relationship between flat minima and generalization.

**Strengths:**

I appreciate the work done by the paper to characterize mitigating factors for when Ansatz 3.1 will not hold (listed at the end of Section 3) and then discussing which of these mitigating factors is at play in certain results.  The results of the experiments are generally clear and cover a wide range of approaches to improve generalization/bias training methods towards flat minima.  Sections 4 and 6 were overall easy to follow.

Note that my lower confidence score is to indicate that I am not as familiar with recent work in this area and that I hope my fellow reviewers can provide additional context on the novelty of this work.

**Weaknesses:**

**Section 5:** I found the logic of this section challenging to follow.  The paper states "Theorem 4.3 tells us that any training procedure that reduces the loss over all data points will also increase the sample-maximum Jacobian norm from a low starting point."  Where does the loss on all data points come into Theorem 4.3?

The results have a number of restrictions (which are mostly acknowledged by the paper) that limit the applicability of the contributions:
* The conditions under which Ansatz 3.1 holds are not rigorously proven.
* The results only hold for simple distributions discussed in Definition 4.1 and 4.2.  The paper gives the example of a GAN with a latent distribution on a hypercube or sphere as an example of a setup where the theory holds, but in most situations this distributional assumption does not hold.
* The experiments are only on CIFAR-10 and CIFAR-100.  Larger scale vision or language tasks are not considered.

Minor Notes:
* Figure 2: I would not use $\lambda$ to denote the weight decay when much of your paper is discussing eigenvalues of the Hessian.  You could just title the $x$-axis weight decay.
* A recent paper to add to the background section on "Flatness, Jacobians, and Generalization." The work does an empirical analysis about the claims on many of the cited papers in larger-scale settings: Maksym Andriushchenko, Francesco Croce, Maximilian Müller, Matthias Hein, Nicolas Flammarion. "A Modern Look at the Relationship between Sharpness and Generalization." https://arxiv.org/abs/2302.07011

**Questions:**

* Would you say that your work essentially points to the sensitivity of the input-output Jacobian being the more fundamental quantity (vs. the sharpness of the loss landscape) when thinking about progressive sharpening and generalization?  Or would you summarize your work a different way?

* Does your work result in any new suggestions for practitioners on what training techniques should be used to best improve generalization. Or put another way, what would you say we gain from the understanding presented in this paper?

---

> ### Author Response · Authors · 2023-11-16
> **Thank you for your time and appreciation of our work**
>
> **Section 5 being challenging to follow** We apologise for the ambiguity. In Theorem 5.1, $\ell$ is just a constant number, such that the loss value at each data point is bounded above by $\alpha\ell^2$ (where $\alpha$ is the coefficient corresponding to the cost function $c$). In Equation (3) we used $\ell$ to denote the empirical loss function of a neural network, which has likely contributed to the ambiguity. We will fix this notation in the next version.
>
> **Restrictions on results**
>
> * **Ansatz 3.1 not rigorous** Making the ansatz rigorous will require making explicit the only implicit (and highly nonlinear) relationship between input-output Jacobian and Gauss-Newton conjugate in Equation 4. This is further complicated by the mediating factors we listed in the second paragraph below Ansatz 3.1. Ultimately we believe that sorting this out rigorously in even a simple setting would constitute an entire paper on its own and is outside the scope of this paper, where we seek only to study these objects qualitatively and show their potential in accounting for unexplained phenomena.
>
> * **Theorem 4.2 holds only for simple distributions** The uniform distribution on the unit sphere has recently been employed to great effect in practical GANs [1], so we do not believe this is a significant restriction. We will include this reference in the next version.
>
> * **Experiments only on CIFAR10/100** Running all of our experiments on larger datasets would have been computationally difficult: already our experiments used approximately 3500 GPU hours (see Computational Resources section in appendix).
>
>
> **Minor comments**
>
> * **Figure 2** Thank you for the recommendation. We will fix that.
>
> * **Missing reference** Thank you for pointing out that relevant paper. We were previously aware of it but had forgotten to include it. We will include it in the new version.
>
> **Would you say that your work essentially points to…** Yes, that is a good summary of our work.
>
> **Does your work result in…** One standout recommendation for practitioners would be to regularise model Jacobian instead of loss sharpness. Our results in Appendix D suggest that doing this even naively can go some of the way to alleviating the poor generalisation associated with small learning rate training. Having said that, sophisticated existing methods (specifically sharpness-aware minimisation) already do an excellent job of regularising model Jacobian (see Fig 5 and Figs 20-27). We believe similarly sophisticated methods targeting model Jacobian would be necessary for state-of-the-art performance.
>
> Another important target audience is theoreticians. The phenomena we explore in this paper are a subject of intense theoretical research, presently concerned mostly with toy models. We hope that our identification of general, architecture-independent mechanisms in this paper will assist in future theoretical research.
>
> [1] Chen et al, SphericGAN: Semi-supervised Hyper-spherical Generative Adversarial
> Networks for Fine-grained Image Synthesis, CVPR22.

---

> > ### Comment · Reviewer_7LFk · 2023-11-22
> > **Response to Authors**
> >
> > I want to thank the authors for their clarifying comments.  Regarding Theorem 5.1, I now understand the relation to the data distribution and would indeed strongly recommend changing the notation so it is not confused with the loss function $\ell$.  Also, I think the conclusions of your work would be made clearer if you included your answer to what you would recommend to practitioners in the paper; it provides a useful summary.

---

### Official Review · Reviewer_L7Vu · 2023-11-03

**Soundness:** 3 good
**Presentation:** 3 good
**Contribution:** 1 poor
**Rating:** 3
**Confidence:** 4

**Summary:**

The paper studies the impact of loss curvature broadly on generalization through the lens of input output Jacobian.

**Strengths:**

On an empirical note, the paper has interesting experiments linking the Jacobian norm and the performance of the models.  The paper is well written and makes clear arguments.

**Weaknesses:**

a) The result of theorem 6.1 suffers terribly from the curse of dimensionality. The authors comment that the data is intrinsically low dimension, however it is not trivial to identify this low-dimensional support and therein lies the challenge to understand generalization in deep learning challenging.

b) The result of theorem 5.1 is also very hard to parse and it is not clear how it is linked to progressive sharpening? Yes, in order to estimate a $f_*$ and starting from low curvature or low Jacobian norm the network has to increase sharpness during training.  The surprising aspect of progressive sharpening is that the sharpness increases despite presence of many minima which are flat. I do not think the theorem can capture multiple minima hence cannot explain progressive sharpness.

**Questions:**

a) Does if always hold that the term in right side of inequality of Eq. (9) in the statement of Theorem 5.1 is always positive.

minor:

It would be helpful to reformulate the statement of ansatz 3.1 in technical terms.

---

> ### Author Response · Authors · 2023-11-15
> **Thank you for your time**
>
> **Curse of dimensionality** While our bound does suffer from the curse of dimensionality, this is at least qualitatively consistent with empirical work showing that generalisation does indeed get harder with increasing intrinsic data dimension [1]. While we agree that it is not trivial to identify the low dimensional support of a given data distribution, it is possible to approximate this support using a generative model. For instance, the distribution generated by a GAN with latent space dimension $d$ has intrinsic dimension bounded above by $d$. One would then have to quantify how well the GAN approximates its target distribution. This is an active research topic [2] which is outside the scope of our paper.
>
> **Theorem 5.1 is hard to parse** You have correctly identified the intuition, namely that in order to fit complex data, the Lipschitz constant of the network must grow from a low starting point. Theorem 5.1 just formalises this intuition: if the loss is below a certain threshold across the data points, then the Lipschitz constant of the network must be at least a certain size. That this increase in Lipschitz constant (or Jacobian norm) implies progressive sharpening is something that we only infer from Ansatz 3.1 and is therefore subject to all the conditions that mediate Ansatz 3.1.
>
> We are not sure we understand your intuition behind why progressive sharpening is surprising. As far as we are aware, gradient descent on a neural net for a sufficiently complex dataset will converge to the sharpest minimum to which its learning rate will allow (at least for the square cost, see e.g. [3]), whether or not there are many flat minima in the landscape. Could you please elaborate on your thoughts regarding progressive sharpening and why you think our proposal falls short of explaining it?
>
> **Positivity of the right hand side of Eq. (9)** No, the right hand side of Eq. (9) need not be positive. For instance, if the dataset consists of only a single class, then the right hand side of Eq. (9) is negative and the bound is vacuous. However, in this setting no sharpening is observed. While this can already be seen to a large extent in our label smoothing experiments (high label smoothing implies the labels are all approximately identical), we are happy to provide additional experimental support for this claim if you feel it would be beneficial.
>
> **Ansatz 3.1 in technical terms** We agree that this is ultimately desirable, but believe that it is necessary to first establish intuition and a qualitative understanding of how the relevant objects behave, which is our aim in this paper. We believe that technically quantifying Ansatz 3.1 at this early stage in our understanding of it would be of little benefit for future research and is outside the scope of the paper.
>
> [1] Pope et al, The Intrinsic Dimension of Images and Its Impact on Learning, ICLR21.
>
> [2] Biau et al, Some Theoretical Insights into Wasserstein GANs, JMLR, vol 22, 2021.
>
> [3] Cohen et al, Gradient Descent on Neural Networks Typically Occurs at the Edge of Stability, ICLR21.

---

### Official Review · Reviewer_LTAo · 2023-11-06

**Soundness:** 2 fair
**Presentation:** 2 fair
**Contribution:** 2 fair
**Rating:** 3
**Confidence:** 4

**Summary:**

This paper explores the relationship between the loss Hessian and the input-output Jacobian via an ansatz, motivated largely based on intuition, by decomposing the Gauss-Newton part of the Hessian into constituting matrices that are also contained in the Jacobian. Some theoretical results are derived as to the extent the maximum input-output Jacobian norm captures the Lipschitz norm of the function as well as its behaviour during training and a simple link to generalization. Overall, these are used to explain the cause of progressive sharpening (related to the edge of stability) as well as some inconsistencies in the flat minima hypothesis.

**Strengths:**

- The paper provides an interesting perspective on how the loss Hessian and input-output Jacobian are closely connected.
- There are some interesting but preliminary results on the behaviour of Hessian maximum eigenvalue (referred to as sharpness) and Jacobian norm in different scenarios, with varying kinds of regularization strategies. It gives an impression that when the low value of sharpness arises due to smaller contributions from the Jacobian, then a lower generalization is implied.
- Understanding when and to what extent the flat minima hypothesis exactly holds is an important research direction. So this work might help towards this end.

**Weaknesses:**

- **The Ansatz is pretty crude and the overall narrative overly simplistic:** The authors themselves admit the lack of rigour in the ansatz, but frankly, there is a lot of hand-waving that is going on throughout the paper. Whatever does not fit the bill is assigned crudely to the other non Jacobian terms in the Gauss-Newton, and that too is done pretty sloppily by just showing the behaviour of individual terms on their own, rather than their interaction which is what matters. Essentially, this lack of even a basic quantitative attribution can be traced to the crudeness of the hypothesis.

&nbsp;

- **Theoretical results are fairly simple and lack any evaluation:** There is nothing significant going on in here that is totally novel; a lot of it is based on past works and the rest seems fairly simple massaging of terms and inequalities here and there. The nature of the theoretical results, like Theorem 5.1, 6.1 which are hard to even evaluate numerically, casts a doubt upon their utility and perhaps suggests even the vacuousness of the bounds. Despite all of this, several statements in the paper sloppily attribute empirical observations to their theoretical results, ignoring the fact that many of the results are just pure one-sided bounds which are insufficient to explain phenomena (like progressive sharpening). Other statements are merely speculative, like that about the exceptional decrease in the Jacobian norm in Figure 4.

&nbsp;

- **Unconvincing empirical results**: The presented experiments are interesting, but fail to be sufficiently convincing.

   - (a) Figure 1 only shows a very short duration in training and when considered over the entire course of training (Figure 29) the relationship becomes flimsy. Even over the initial short duration, this relation is not as clean, as can be seen in Figures 8 - 12. Further, scales of the Jacobian norm and that of Hessian sharpness get pretty far apart for non-zero label smoothing.

    - (b) Likewise the particular curves of the generalization gap and the sharpness or Jacobian in Figures 2 and 5 do not give a firm support for their claims. Comparing results across different learning rate values does not give a clear picture of the trend (e.g., looking at fixed values of weight decay in Figure 2, one would doubt if sharpness and Jacobian are even sufficiently capturing generalization). This makes me wonder about what are the actual correlation coefficients rather than just seeing a rough pictorial correlation. Similarly, the Jacobian norm increase midway in Figure 5 also seems a bit weird and the actual correlation seems unclear. Except for the case of the lowest learning rate, the rest of the scenarios are not convincing --- but then one of the selling points of the paper was explaining the generalization benefits of an initially large learning rate!

   - (c) Then the batch size experiments have probably been run for a fixed number of epochs, resulting in less updates when a bigger batches are used, and thus could be a potential confounder in the results. What happens when compared across number of updates, instead of epochs on the x-axis? Besides, it's good that the training loss is shown, but it would make more sense to compare the gradient norm of the loss to compare their relative extents of convergence.

&nbsp;

- **Literature on Jacobian norm:** The paper does not fully demarcate their contributions from that of Gamba et al 2023. Also, the discussion of Jacobian norms in (Khromov & Singh, 2023; https://arxiv.org/pdf/2302.10886.pdf), their relation to generalization, and the bound on the variance via the Lipschitz, bear similarities to the some of the material presented here.

**Questions:**

^^

---

> ### Author Response · Authors · 2023-11-15
> **Thank you for your time**
>
> **Lack of quantitative evaluation, looseness of bounds** While we agree that a tight, quantitative analysis is ultimately desirable, we are not presently aware of any works which are able to provide such an analysis outside of toy problems. Performing such tight quantitative analysis in the practical settings we consider is impossible without first identifying the correct theoretical objects to study and giving a qualitative account and empirical validation of their behaviour in practical settings. This is what our work proposes to do, and we are aware of no others which do this in the context we are considering. To reject the paper on the grounds that it does not exhibit a complete, quantitatively tight analysis in practical settings would, we believe, be self-defeating given the current state of knowledge.
>
> **The Ansatz is pretty crude and the overall narrative overly simplistic**: It seems you’re misunderstanding our purposes behind this analysis, especially in relation to the literature. “Whatever does not fit the bill” are known failure modes of the flat minima hypothesis and progressive sharpening, which to date have not been accounted for even qualitatively (see the second paragraph following Ansatz 3.1). Our proposal gives accounts for these diverse failure modes in terms of a single common theoretical object (the conjugate of the Gauss-Newton matrix). Far from being inconveniences to a simplistic narrative, these failure modes and the explanations we give for them point to the subtlety of the mechanism we are proposing and its potential as a target for future research.
>
> The “interactions” you mentioned sound intriguing, and we would appreciate if you could expand on them. Please note though that any one of the qualitative experiments we ran in analysing the failure modes of the ansatz could easily have falsified our proposal (for instance, if in Figures 30 and 31 the parameter derivatives did not scale oppositely to the Jacobians as the data is scaled, with sharpness nonetheless still behaving oppositely). Our experiments therefore do "matter", and provide nontrivial support for our arguments.
>
>  **Theoretical results are fairly simple**: The researchers our work is targeted towards are those who are seeking a greater understanding of deep learning. If we are able to provide such understanding with simple proofs, so much the better.
>
> **Unconvincing empirical results**:
>
> (a) Figure 1 shows training to 99% train accuracy with *full batch* gradient descent, following [1]. Figure 29 depicts training with SGD, and its deviation from Figure 1 is precisely the point of that section: the relationship breaks down later in training, as you have pointed out. Of Figures 8-12, the only one that seems noisy is Figure 9: keep in mind that these are randomly initialised networks with only 5 trials, so outliers have an outsized effect and some noise is to be expected. The large scaling difference between Hessian and Jacobian norm does not run counter to any of our arguments and need not be surprising given the nonlinear, implicit relationship we are positing between them in Equation 4. Quantitative explication of this scaling is outside the scope of our already lengthy study.
>
> (b) The trend evident in Figure 2 is that across all learning rates, Jacobian correlates positively with generalisation gap while Hessian correlates negatively, as weight-decay is increased. That is all that the experiment was designed to test. While for fixed weight-decay the relationship across learning rates is indeed unclear, we do not claim that Jacobian norm is the only factor in generalisation (see the variation term in Theorem 6.1) and are instead arguing that loss flatness correlates with generalisation only insofar as it is associated with smaller Jacobian norm. Regarding the Jacobian increase in Figure 5, please see the caption for a possible explanation. Even with this behaviour, the Jacobian correlates more consistently with generalisation than sharpness does, which is all we are claiming in that figure.
>
> (c) We took this consideration into account by terminating training only after the loss was sufficiently small on average over a uniform selection of a number of previous iterations. Please see Appendix B.2 for complete details.
>
> **Literature on Jacobian norm**: Both Gamba et al (G) and Kromov et al (K) study Lipschitz properties of neural nets via their Jacobians as we do, but with different purposes. Both G and K are concerned with double descent, whereas we are not. Neither G nor K attempts to give an account of progressive sharpening, and neither attempts to give a PAC generalisation bound, whereas we give both. Neither G nor K give theorems on how the Jacobian approximation to the Lipschitz constant scales with number of training points, whereas we do. We will update the paper to cite K as it is indeed relevant.
>
> [1] Cohen et al, Gradient Descent on Neural Networks Typically Occurs at the Edge of Stability, ICLR21

---

### Public Comment · ~Zhiwei_Jia1 · 2023-11-23
**Related Work**

Hi authors. I like your work and think it would be great to add the following work [1], which also utilizes the spectrum of the DNN loss Hessians for measuring generalization, to the related work. Thanks.

[1] Information-Theoretic Local Minima Characterization and Regularization

---

### Meta-Review · Area_Chair_eQeY · 2023-12-05

**Metareview:**

The paper is aimed at contributing to a line of work investigating connections between the curvature of the loss landscape (here measured by the Hessian eigenvalues) and the input-output Jacobian. The authors focus on settings in which the two are assumed to vary together, an assumption referred to in the paper by ansatz 3.1, and provide several theoretical results concerning two topics in the theory of DL, namely, progressive sharpening and generalization properties of flat minima.

The reviewers appreciated the perspective taken in the paper based on ansatz 3.1: its scope as well as its implications to loss curvature vs. input-output model behavior, and generally found the numerical work linking the Jacobian norm and the performance of models interesting. However, several critical issues were raised by the reviewers, primarily concerning the validity and the relevance of results not covered by previous work to curvature and learning. Initial evaluations remained consistent during the rebuttal phase. The authors are encouraged to incorporate the important feedback given by the knowledgeable reviewers.

**Justification For Why Not Higher Score:**

Incoherent story

**Justification For Why Not Lower Score:**

N\A

---

### Decision · Program_Chairs · 2024-01-16

Reject